

# Enhancing the performance of the aggregated bit vector algorithm in network packet classification using GPU

Mahdi Abbasi[1], Razieh Tahouri[2] and Milad Rafiee[1]

[1] Department of Computer Engineering, Engineering Faculty, Bu-Ali Sina University, Hamedan, Iran
[2] Department of Computer Engineering, Engineering Faculty, Islamic Azad University of Hamedan, Hamedan, Iran

## ABSTRACT

Packet classification is a computationally intensive, highly parallelizable task in many advanced network systems like high-speed routers and firewalls that enable different functionalities through discriminating incoming traffic. Recently, graphics processing units (GPUs) have been exploited as efficient accelerators for parallel implementation of software classifiers. The aggregated bit vector is a highly parallelizable packet classification algorithm. In this work, first we present a parallel kernel for running this algorithm on GPUs. Next, we adapt an asymptotic analysis method which predicts any empirical result of the proposed kernel. Experimental results not only confirm the efficiency of the proposed parallel kernel but also reveal the accuracy of the analysis method in predicting important trends in experimental results.

# INTRODUCTION

The considerable evolution in the speed of internet communications makes the gap between communication speed and processing speed ever wider. To resolve this problem, recent network systems have deployed flow-based traffic processing instead of packet-based processing. For this purpose, the packet classification technology is used as a fundamental process in their architecture. In packet classification, arrival packets are divided into distinctive flows based on a set of predefined filters. Then, the same actions could be applied on batches of packets in the same flows.

To classify an incoming packet, the content of certain fields from its header are matched against corresponding conditions in the filters which have been statistically or dynamically defined. The classifier filters are of different priorities. Therefore, if a packet matches more than one filter, the packet is tagged by the label of the stream corresponding to the filter which is of the highest priority (*Katsikas et al., 2016*; *Taylor, 2005*).

The algorithms of packet classification are implemented through both hardware and software. Hardware methods have achieved the highest speeds in classification by utilizing parallel lookups on ternary content addressable memories (*Sun et al., 2017*). However, problems such as considerable prices of these hardware modules, their high consumption power, the inflexibility of their architecture to any variation in the filters, and inefficiency

Corresponding author
Mahdi Abbasi, abbasi@basu.ac.ir

in using storage spaces have motivated many kinds of research in the field of the software implementation of packet classifiers. In contrast, the algorithms of software packet classifiers are developed easily and with low costs. In addition, they are highly extensible and flexible. An important disadvantage of such classifiers, however, is their low speed. Therefore, the challenge of accelerating software classifiers of IP packets has become a motivation for many recent studies.

The algorithms of packet classification are categorized into linear search, decision tree, tuple space, and decomposition. Among these, decision tree and linear search algorithms do not have the appropriate structure to be parallelized. The main reason is the great degree of data and control dependence which exists in these algorithms. In contrast, the reduced dependence of tuple space and decomposition algorithms on data and control makes them more appropriate for the parallelism on multi-core and many-core systems. Accordingly, numerous studies have been conducted with the aim of parallel implementation of packet classification algorithms on multicore and many-core machines. Interestingly, a growing body of literature has been devoted in recent years to parallel implementations on graphics processing units (GPUs), especially using GPGPU technologies (*Fan, Xu & Zhao, 2017*). So far, however, no comprehensive analysis method has been proposed for predicting the empirical performance of parallel implementation of these sets of algorithms given the important characteristics of GPUs such as their complex memory subsystem.

Review of the related literature shows that none of the studies have parallelized aggregated bit vector (ABV) algorithm on GPU-like many-core machines. This algorithm is a decomposition-based algorithm and has an appropriate structure that lets it be highly parallelized on GPU systems (*Baboescu & Varghese, 2001*). In this paper, we seek to parallelize this algorithm on the GPU. The other contributions of the paper are as follows:

1. In order to assess the influence of the machine parameters, such as memory latency, shared memory size and the number of allowable threads on the overall performance, the time and memory complexity of the proposed kernel is computed analytically and then compared with the experimental results. The analysis of empirical results and their analytical performance models indicates that the proposed model can help predict the accurate performance of parallel kernels.

2. The evaluation of the experimental results shows that the throughput of the parallel kernel is about 66.54 times more than the throughput of the sequential version of the algorithm. This corresponds to a speedup of about 100 times.

The rest of this article is organized as follows. In section two, the structure of the GPU and the operation method of the ABV algorithm are explained. In the third section, the related works on parallelizing packet classification algorithms on GPU are reviewed. In the fourth section, the proposed parallel kernel of the bit-vector algorithm is described. Then, in the fifth section, analysis of the computing and memory complexity of the suggested kernel is presented and the results are compared with the experimental results. Finally, conclusions and directions for future research are discussed in the last section.

## BACKGROUND

In this section, first the structure of GPU and its memory is studied. Then, the method of constructing the ABV corresponding to a filter set and the algorithm of classifying IP packets using that construct are explained.

### Graphics processing unit

Graphics processing unit is a special system to display graphic images in personal computers. Following the release of software development packages on this unit by great manufactures such as Nvidia (*Nvidia, 2017*; *Nakano, 2013a*) and ATI, the use of GPU was accepted as a powerful parallel computing unit along with central processing unit in accelerating computational processes. The main reason for this big computing revolution is that the architecture of GPU is specially designed for running compute-intensive and parallel operations. Accordingly, Nvidia supplied a software platform called compute unified device architecture (CUDA) for performing nongraphic computations on graphic processors in 2006 (*Nakano, 2013a*) CUDA provides possibilities that could be used by programmers to have access to hardware capabilities of graphic processors in their nongraphic programs and increase the speed of performing complicated algorithms.

Several investigations including *Li et al. (2013)* and *Lin et al. (2016)* have attempted to study the use of CUDA platform for parallel implementation of network functions such as IP lookup in routing tables, aiming at having access to higher throughput. This tool has been used for executing parallel genetic algorithms (*Zhao et al., 2018*), neural networks (*Gong et al., 2017*), and ant colony optimization algorithm (*Llanes et al., 2016*). Also, the capacity of parallel programming in CUDA platform has been used in the field of cryptography for compressing databases and accelerating encryption algorithms (*Przymus & Kaczmarski, 2014*; *Vasiliadis et al., 2014*).

From a programming perspective, two CUDA processes are involved in parallel computations: host and device processes. The former runs on central processing unit and, in fact, executes the main program whereas the latter is executed on GPU. Any program that is written on CUDA may be formed of several kernels. Each kernel is executed by a grid and each grid may be formed of several blocks. Each block is formed of several threads. Indeed, threads are responsible for performing programs.

The graphic processor used in this paper is GT730 that is comprised of two streaming multiprocessors (SMs), each consisting of 192 streaming processors (*Cheng, Grossman & McKercher, 2014*). Figure 1 shows the hardware structure of an Nvidia GPU. Each GPU consists of different types of memory including global, constant, texture, register, and shared memories. The CUDA grid in this figure includes four blocks each of which consists of four threads. In our experiments, two blocks are defined for every SM and 1,024 threads are defined in each CUDA block. Therefore, 4,096 threads are used.

### The ABV algorithm

Aggregated bit vector algorithm is based on decomposition. To construct the data structure of this algorithm, source, and destination IP addresses of filters are used to construct two corresponding binary trees. For example, consider the two-dimensional

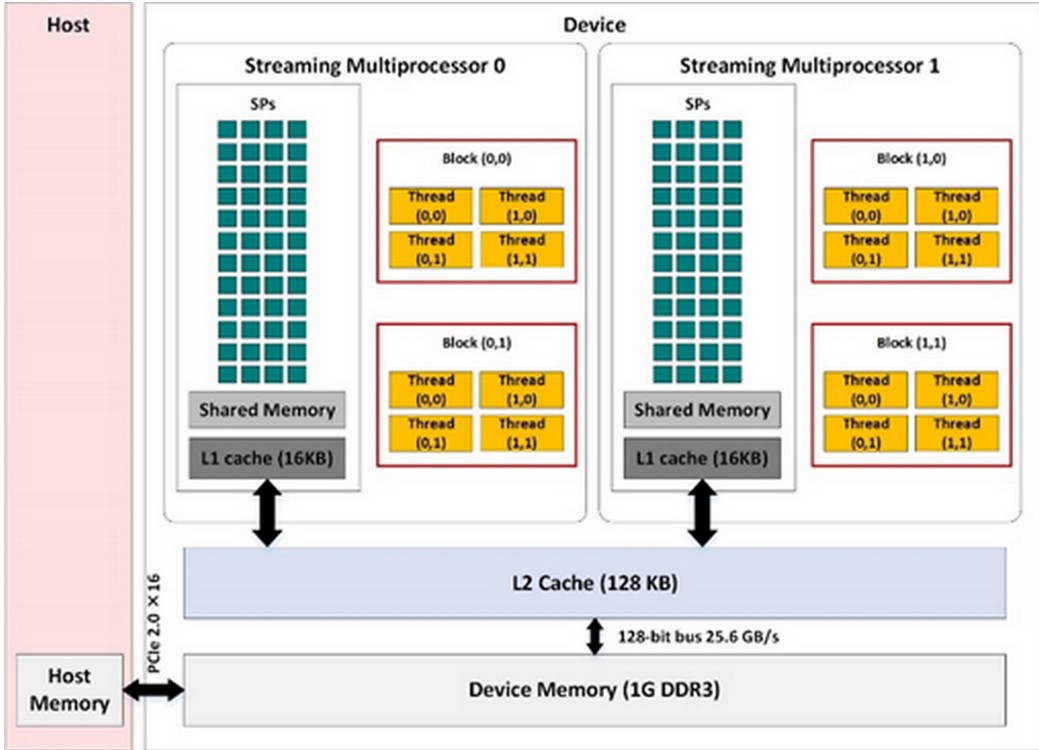

**Figure 1 Architecture of an Nvidia GPU.** Main parts of an Nvidia GPU are streaming multiprocessors and memory hierarchies. Each streaming multiprocessor schedules the running threads of associated blocks on streaming processors.

**Table 1 Sample filter set.**

| Filter | Source IP | Destination IP |
|---|---|---|
| F0 | 00* | 00* |
| F1 | 00* | 01* |
| F2 | 10* | 11* |
| F3 | 11* | 10* |
| F4 | 0* | 10* |
| F5 | 0* | 11* |
| F6 | 0* | 0* |
| F7 | 1* | 01* |
| F8 | 1* | 0* |
| F9 | 11* | 0* |
| F10 | 10* | 10* |

**Note:**
Columns from left: filter name, source IP (or $F1$ field) and destination IP (or $F2$ field).

filters existing in Table 1. For each dimension of the filter set, a binary tree has been constructed and shown in Fig. 2. Note that, the priority of filters descends from top to bottom in Table 1. To form the source tree, the source IP address of the filter is read bit by bit. Then, the left or right side of the tree is traversed corresponding to each bit "0" or "1," respectively. This movement is repeated considering each consecutive bit of the

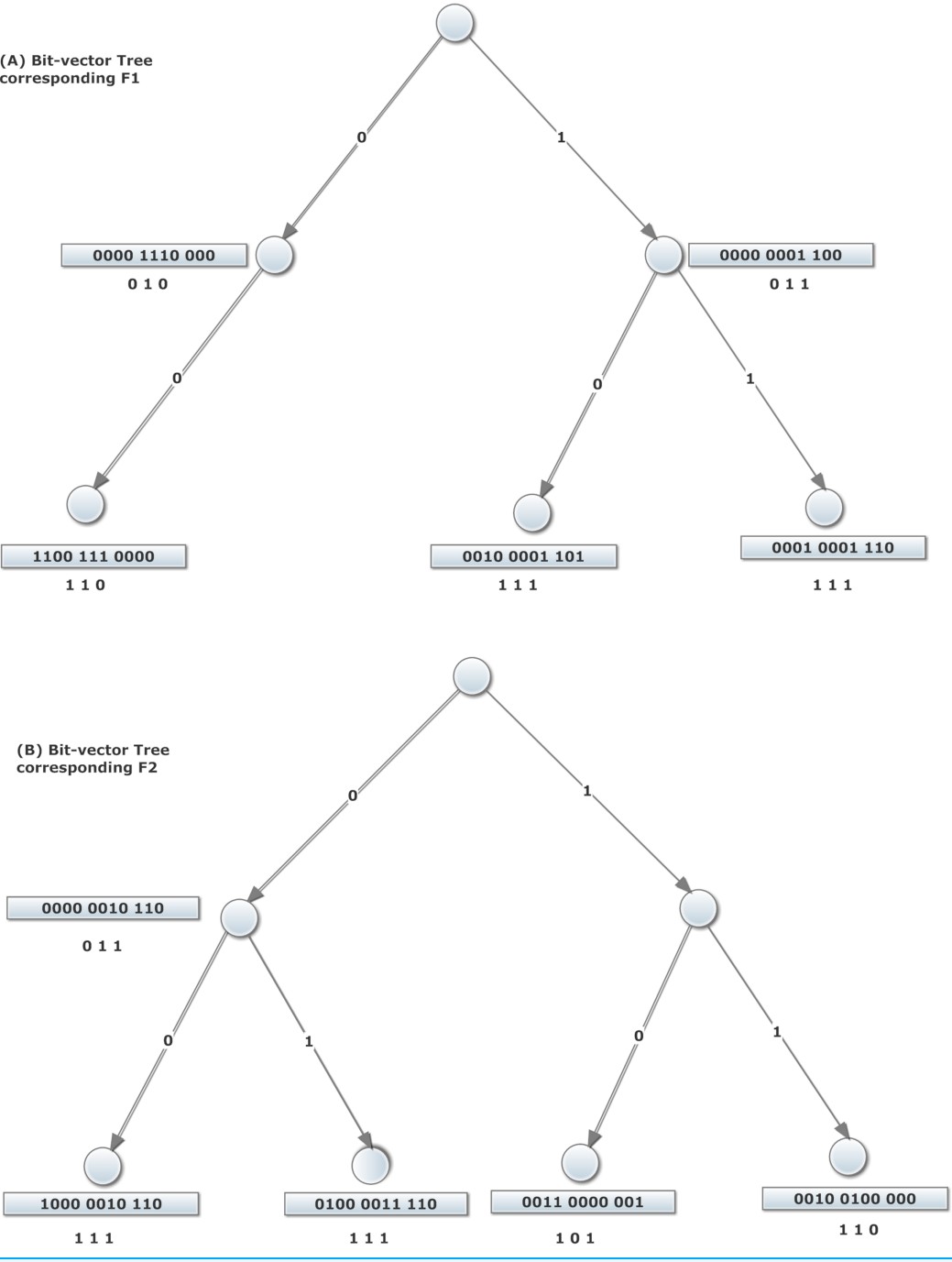

**Figure 2 ABV algorithm structure with aggregation size *A* = 4, corresponding to the filters of Table 1.** The trees illustrated in the top and bottom correspond to bit-vector trees of *F*1 and *F*2 fields of the filters of Table 1, respectively. For each node of these trees, the corresponding bit-vector is depicted close to it.

source IP prefix of the filter and is terminated by the last bit of it. At this point, the node corresponding with the mentioned prefix is established. Each node in the tree for a field is labeled with an *N*-bit vector. Bit *j* in the vector is set if the prefix corresponding to rule $F_j$ in the filter set matches the prefix corresponding to the node (*Baboescu & Varghese, 2001*).

In Fig. 2, given the match of prefix 00* in Field 1 by the values 00* and 0*, which correspond to values in filters 1, 2, 3, 7, and 8, the 8-bit vector corresponding to leftmost leaf node in the right trie of Fig. 2 is 11100011. In order to construct the ABV, the bit vector of each node is divided into A-bit parts. In any A-bit aggregation, if at least one of the bits equals one, the aggregated bit is set; otherwise, it is reset. Similarly, the tree of destination IP addresses of filters is constructed in the same way. The router which uses ABV for forwarding a packet, should perform a longest-prefix match for each distinct field $H_i$ of the packet header, in the corresponding trie $T_i$. The result of this match is a trie node $N_i$. As mentioned above, each node contains both the bit vector and the ABV. Next, the *AND* of ABVs is computed. If the value of the position $j$ is 1 in the *AND* of the aggregate bit vectors, an *AND* operation is done on the corresponding parts of bit vectors (bits $A \times j, \ldots, A \times (j+1) - 1$). A filter which its corresponding bit in *AND* result is set, is a matched filter. Finally, among all matched filters, the algorithm selects the filter with highest priority (lowest index) as the best matched filter. For example, Fig. 2 demonstrates the construction of the ABV structure corresponding to the example database in Table 1. Let us consider a packet with (Field1, Field2) = (0010, 0100). From Fig. 2, it is clear that the longest-prefix matches for these two fields are 00 and 01, respectively. The related 11-bit regular vectors are 11001110000 and 01000011110, respectively. Here, given an aggregation size $A = 4$, the related ABVs (shown below the regular bit vectors) are 110 and 111, respectively. Since the *AND* of these two aggregated vectors produces 110, we should only examine the first eight filters of filter set to find the possible matching filters. This eliminates the need to examine the remaining four filters. In addition, given the priority of filters, one can examine the first quarter of source bit vector 1100 and destination bit vectors 0100. The logical *AND* of this two 4-bit yields 0100. Consequently, since the second bit of the result is 1, the best matched filter would be *F*1.

When several filters with various priorities match the information of the incoming packet header during traversal, the filter is selected as the output which is of the highest priority.

## RELATED WORK

Graphics processing unit-based packet classification is one of the many parallel fields in the literature that have emerged after the introduction of GPU for parallel processing. Since the emergence of the great capability of GPUs in performing parallel algorithms, a few studies have been conducted on parallel execution of packet classification algorithms on such powerful many-core machines. A seminal study in this area is the work of *Nottingham & Irwin (2009)* which theoretically investigates the feasibility of parallel execution of packet classification algorithms on GPUs. They introduce the concept of parallel packet classification on CUDA and OpenCL platforms. However, their work lacks implementation and effective evaluation. *Hung et al. (2011)* evaluate two parallel packet classification algorithms, namely BPF and RFC, with eight and four different scenarios of using GPU memory, respectively. According to their empirical results, for both algorithms, the most efficient scenario is one in which the classifier filters and packets are stored in

constant and global memory, respectively. The main weakness of their study is that they use an unrealistic dataset, which includes only three filters as the classifier filter set. Certainly, the empirical results of such an unrealistic model may not be sufficiently reliable.

*Deng et al. (2011)* propose a hybrid microarchitecture by integrating both CPU and GPU for parallelizing linear packet classification algorithm. In this study, only the slow global memory is used. Consequently, despite the reported dramatic acceleration, their hybrid microarchitecture may not be efficient in accurately exploiting the capabilities of the memory subsystem of GPUs. *Kang & Deng (2011)* propose the idea of meta-program for GPU-based packet classification. This technique compiles the rules into instructions to avoid the expensive latency of memory accesses, as claimed by the authors. The authors offer no explanation about their data layout in the memory subsystem of GPU. A more comprehensive study would examine the effect of storing such meta-programs in different memory modules of GPU on the overall performance of parallel algorithms.

*Zhou, Singapura & Prasanna (2014)* addresses the impact of efficiently using powerful parallelism and various types of memory on the performance of algorithms such as packet classification on many-core machines. They investigate GPU's characteristics in parallelism and memory accessing, and implement a Bit-vector packet classifier using CUDA. Their parallel classification kernel exploits only one CUDA block with 32 threads (one warp). In this kernel, classifying each packet involves two phases. In the first phase, each of the 32 threads examines all required fields of $N/32$ filters sequentially and produces a local classification result. Then, in the second phase, the filter with the highest priority among the 32 local results is identified in five steps of comparison. In their recommended data layout, the shared memory of the block is divided into 32 equally-sized memory banks. Each bank is filled with the same set of filters. The remaining filters and all required bit vectors are stored in the global memory of GPU. The idea behind this type of mapping is to minimize the chances of memory bank conflict as much as possible.

However, the performance of this kernel declines significantly as the number of filters increases. In such a case, the shared memory cannot hold the entire dataset and the remaining data is stored in the slow global memory of GPU. Consequently, the access time of threads increases dramatically.

In their groundbreaking study, *Varvello et al. (2016)* proposed a more effective parallel kernel for classifying packets on GPUs on the basis of the characteristics of their memory subsystem. The kernel of this parallel model is designed to maximize parallelism by splitting the filter set among several blocks so that each block is responsible for checking the incoming packets only against part of the filter set. The authors demonstrate the effect of some parameters including size of the filter set, number of blocks as well as threads per block on the overall efficiency of the parallel kernel of packet classification. The study would have been more comprehensive if the authors had adopted an efficient analysis method to predict the effect of different characteristics of GPU and algorithm on the overall performance and explain all the empirical results.

*Zheng et al. (2015)* experimented with Hierarchical Intelligent Cuts algorithm on GPU with different numbers of threads and blocks. In their proposed model for kernel, each

thread is responsible for classifying a certain number of packets. *Qu et al. (2015)* experimented with bit-vector algorithm on a hybrid GPU-CPU system. In their proposed kernel, on the coming of a new packet, five GPU threads are exploited, each of which classifies the packet based on the value of each field and creates a bit vector. Next, the CPU combines all bit vectors, which correspond to different fields of the packet in order to find the best-matched filter. They show that reduction of four parameters including the delay of transferring data from host to device, the time required for the kernel search, the delay of transferring data from device to host, and the time required for combining results enhances the performance.

These studies clearly indicate that research in the field of GPU-based packet classification has not yet considered ABV algorithm. The reviewed studies highlight the need for parallelizing ABV algorithm and exploiting a comprehensive analysis model to predict its performance on GPU. Constructing such an analysis model for GPU-based packet classification requires knowledge of system characteristics as well as the algorithm's nature.

In the next section, we explore our practical model for GPU-based classification of IP packets according to ABV, which is enriched using TMM model.

## PARALLEL IMPLEMENTATION OF ABV ALGORITHM

Several stages are needed for parallel execution of the aggregated bit-vector algorithm on GPU. In the first step, the filter set is used to construct the corresponding trees which are required in the aggregate bit-vector algorithm. Then the structures of the tree, filters, headers of the packets and array for storing the results of classification are copied from the host memory into the global memory of the graphic processor. After this step, the grid structure is specified in CUDA by defining the number of blocks and the number of corresponding threads in each block. Next, a predefined scenario is followed which specifies how the threads can pick the packets and classify them concurrently. At the end of scenario, the result matrix which contains the identification of rules which are best matched with packets is transferred to the host memory.

After loading the filters of the filter set, their source and destination IP addresses are used to establish the required trees based on the bit-vector algorithm. Given the pre-specified aggregation size, the ABVs which correspond to the bit vectors of nodes are computed and stored in those nodes. In this paper, the aggregation size is taken $A = 10$. All this is considered as the pre-processing step of the algorithm.

After this pre-processing, the number of blocks and threads in each block is specified. Now, it is the time to classify the packet by the threads. In the following, we will explain all of the above steps according to the pseudo-code of Fig. 3. According to the pseudo-code, the global memory of GPU is used for maintaining the required data structure of the algorithm. This memory module has enough capacity to hold all of the required data set. Hence, all threads of different blocks have access to the data laid in it. Given the number of the described threads in GPU, each thread is responsible for classifying a definite number of packets. Each thread can be distinguished from other threads using its specific index in the block and the block index in the grid. As indicated in line 4, a unique

```
Input: filters F, tree Src_IP_Tree, tree Dst_IP_Tree, headers H
Output: Reslut_Array
        Pre-Processing://performed within host (CPU)
 1:  Construct Src_IP_Tree and Dst_IP_Tree from F
     Transfer: // from host to global memory of GPU
 2:    GPU Global memory ← CPU memory(Src_IP_Tree, Dst_IP_Tree, H, F, Result_Array)
```

$$3: \quad \text{for } i = 0 \text{ to } i < \frac{|H|}{|Blocks| \times |Threads\ in\ Block|} \text{ do}$$

$$4: \quad ThreadIndex \leftarrow threadIdx + i \times (|Blocks| \times |Threads\ in\ Block|)$$
$$+ (blockIdx \times |Threads\ in\ Block|)$$

```
 5:      packet ← ReadPacket(ThreadIndex)
 6:      Src_ABV ← Classify(Packet, Src_IP_Tree)
 7:      Dst_ABV ← Classify(Packet, Dst_IP_Tree)
 8:      ABV ← AND(Src_ABV, Dst_ABV)
 9:      BMF ← Check ABV bits and Match(Other Fields)
10:      if BMF ≠ Null then
11:          Result_Array(Packet_ThreadIndex, BMF)
12:      end if
13:      i ← i + 1
14:  end for
     Transfer: // from global memory of GPU to host
15:  CPU memory ← GPU Global memory (Result_Array)
```

**Figure 3 ABV pseudo-code.** The pseudo-code of the proposed parallel ABV algorithm includes pre-processing and processing (transferring data from the system memory to the device memory, classifying packets concurrently, and finally transferring the results to the host memory).

*Thread Index* is computed for each thread. This number is used for specifying the packets which should be classified by that thread.

The method to classify the packets by the threads is so that each thread chooses a packet from the global memory of the device (line 5). Then, it performs the classification operation based on the source and destination bit-vector trees which exist in the global memory of the device. For this purpose, each packet is classified by traversing according to its source/destination IP address on corresponding bit-vector trees (lines 6–7). Then, the logical AND operation is performed on two ABVs which were achieved in the traversal (line 8). Now, corresponding to any set bit in the resultant ABV, its corresponding parts in the main bit vectors are examined from left to right. In this detailed examination, the filters whose corresponding bits in both source and destination bit vectors are set are tested to find the filter with the highest priority that matches completely with the intended packet (line 9). For this purpose, other fields of packet are inspected linearly. If the result of classification is not null, the index of the best-matched filter would be stored in the proper index of *Result_Array* (lines 10–12). This operation is repeated for all packets.

Note that, to find the filter which best matches an input packet, it is necessary to compare other fields of it with the corresponding fields of the candidate filters (line 9). These fields include source port number, destination port number, and protocol number.

Therefore, it may be required to compare the value of three fields of the packet with the corresponding fields in all candidate filters. Regarding the considerable latency of global memory, a trick is required to accelerate this process. We increase the speed of this step by transferring the fields of the packet header to the local memory of the block (line 5). Index of the filter which is best matched with the corresponding packet is stored in *Result_Array*. This vector is transferred to the host memory as the final result of packets classification (line 15). Finally, the occupied memory is returned to the system.

## COMPLEXITY ANALYSIS

Computational and memory complexity computing is one of the important tools for predicting and analyzing the efficiency of algorithms. Different analytical models have been presented for studying the efficiency of parallelized algorithms in the platform of graphic processors. Some models like TMM (*Ma, Agrawal & Chamberlain, 2014a*), PGM (*Kirtzic, Daescu & Richardson, 2012*), BSP (*Amarōs et al., 2015*), HMM (*Nakano, 2013b*), DMM (*Nakano, 2014*), UMM (*Nakano, 2014*), and MMM (*Haque, Maza & Xie, 2014*) function asymptotically; in other words, in addition to the characteristics of parallel algorithm, these models consider some of the main characteristics associated with the architecture of graphic processor in computing memory and computational complexities.

In contrast, some other analytical models called calibrated models address such details in their efficiency analysis that are not important in asymptotic models, thereby analyzing the complexity and approximating the efficiency of parallel algorithms on graphics processors more accurately. For example, the calibrated method is used in the model presented by *Hong & Kim (2009)* or the model presented by *Liu, Müller-Wittig & Schmidt (2007)*. The need for deep knowledge about the hardware details of the graphics processor is among the problems of this model in having access to some of the required parameters of analysis. Another problem of this model is that it does not consider the effect of some parameters such as the hit rate of the cache memory of graphics processors.

A new model by *Ma, Chamberlain & Agrawal (2014b)* has recently been suggested for analyzing the complexities of parallel algorithms on graphics processors. This model is obtained from the combination of asymptotic and calibrated models. In this model, in addition to asymptotic analysis, parameters such as the sequential processing time of the algorithm, the number of processing cores, the number of transfers to/from memory and the number of threads in each core are considered. In this paper, this model is adapted for analyzing the performance of packet classification algorithms. The required parameters for this analysis model are presented in Table 2.

In the combined model of analysis, the total time of executing ABV algorithm on the CPU is obtained by Eq. (1) as:

$$T_1 = a \times 2 \times \left( O(\log N) + O\left( \left\lceil \frac{N}{A^k W} \right\rceil \right) + O(N) \right) \tag{1}$$

According to our experimental setting, the parameters of Table 3 could be considered in analytical computations. Consequently, the total time of executing the algorithm on the

**Table 2 TMM model parameters.**

| Parameter | Description |
|---|---|
| $Q$ | Number of cores per core group (SMs) |
| $L$ | Time for a slow global memory access |
| $P$ | Total number of processors (cores) |
| $T_1$ | Total number of operations in the program (work) |
| $M$ | Number of global memory transactions |
| $\tau$ | Number of threads per core |
| $n$ | Number of filters in classifier |
| $a$ | Number of packets |
| $B_a$ | Number of active thread blocks on each SM |
| $B_r$ | Number of requested thread blocks for run parallel algorithm |
| $n_T$ | Number of threads on each block |
| $n_r$ | Number of access to memory for register the result |

Note:
   The parameters required for the analytical prediction of the complexity of the parallel ABV algorithm are defined in each row.

**Table 3 Parameters used for complexity computation of ABV algorithm.**

| Parameter | Notation/ formula | Value |
|---|---|---|
| Aggregate size | $A$ | 10 |
| Memory word size in Byte | $W$ | 4 |
| Number of GPU cores | $P$ | 384 |
| Number of threads | $Threads$ | 3,072 |
| GPU global memory access time | $L$ | 100 |
| Number of threads per core | $\tau$ | 8 |
| Memory access width | $C$ | 32 |
| Span of ABV | $2 \times O(\log N) + \frac{N}{A}$ | |
| Complexity of search on ABV | $O(\log N)$ | |
| Computational complexity of merging search results in $T_1$ | $O\left(\left\lceil \frac{N}{A^k W} \right\rceil\right)$ | |
| Computational complexity of merging search results in $T_p$ | $O\left(\left\lceil \frac{N}{A^k C} \right\rceil\right)$ | |
| Complexity of finding best matched filter | $O(N)$ | |

Note:
   The parameters of the TMM model are computed using the specifications of GPU and ABV algorithm and then inserted in the table.

graphic processor for the proposed kernel models and its memory complexity are obtained by Eqs. (2) and (3), respectively:

$$T_P = O\left(\max\left(\frac{T_1}{P}, T_\infty, \left\lceil \frac{a}{\tau \times P} \right\rceil \times M \times L\right)\right)$$

$$= O\left(\max\left(\frac{a \times 2 \times \left(O(\log N) + O\left(\left\lceil \frac{N}{A^k W} \right\rceil\right)\right) + O(N)}{384}, 2 \times O(\log N) + \frac{N}{A}, \left\lceil \frac{a}{8 \times 384} \right\rceil \times 2 \times \left(O(\log N) + O\left(\left\lceil \frac{N}{A^k C} \right\rceil\right) + O(N)\right) \times 100\right)\right)$$

(2)

**Table 4 System characteristics.**

| Characteristics | | |
|---|---|---|
| CPU | | Intel Corei5-6400T@2.2GHZ |
| RAM | | 4G |
| Operating system | | Windows8,64bit |
| GPU | Model | GeForce GT730 |
| | Cores | 384 |
| | SMs | 2 |
| | Max number of threads per block | 1,024 |
| | Bus width (bit) | 64 |
| | Clock rate memory | 900 MHz |

**Note:**
The specification of the CPU and the GPU of the system which was used in experiments are illustrated.

$$M = 2 \times \left( O(\log N) + O\left( \left\lceil \frac{N}{A^k C} \right\rceil \right) + O(N) \right) \tag{3}$$

In order to assess the accuracy of the predicted complexities we have conducted some experiments. In the following, after explaining the classification experiments on GPU as well as CPU, the results are analyzed.

## Implementation and evaluation

In this section, first ClassBench is introduced. This tool is used for producing synthetic filter sets and synthetic packet headers. Then, the hardware specification of the computer system and the GPU device which were used for executing the proposed kernel code is explained. Afterward, the results of implementing the serial and parallel versions of ABV algorithm on the synthetic dataset are analyzed and evaluated from the efficiency perspective.

## ClassBench

ClassBench is a simulator for producing synthetic filters with desired distributions in the geometric space of filters. This tool produces dummy packets corresponding to the produced filters. Indeed, it creates filters with distributive parameters that are given to it as input. The presence of this simulator satisfies the need for real and heterogeneous filters of Firewalls, IP-Chains, and Access Control Lists. In the majority of the studies (*Deng et al., 2011*; *Kang & Deng, 2011*; *Varvello et al., 2016*; *Zheng et al., 2015*; *Zhou, Singapura & Prasanna, 2014*), ClassBench has been used for producing the required data structure due to a need for filters and packets that are close to reality in terms of structural characteristics and statistical distribution. In this study, this tool was used to produce the set of filters corresponding to ACL parameter containing 1K, 2K, 4K, 8K, and 10K of filters along with 1K, 4K, 32K, and 256K of packets for evaluation of the kernel.

## System setting

The proposed kernel has been implemented in a system with characteristics mentioned in Table 4. In this implementation, the CUDA programming framework version 7 based on C++ language is used. In this study, we have used the whole capability of the

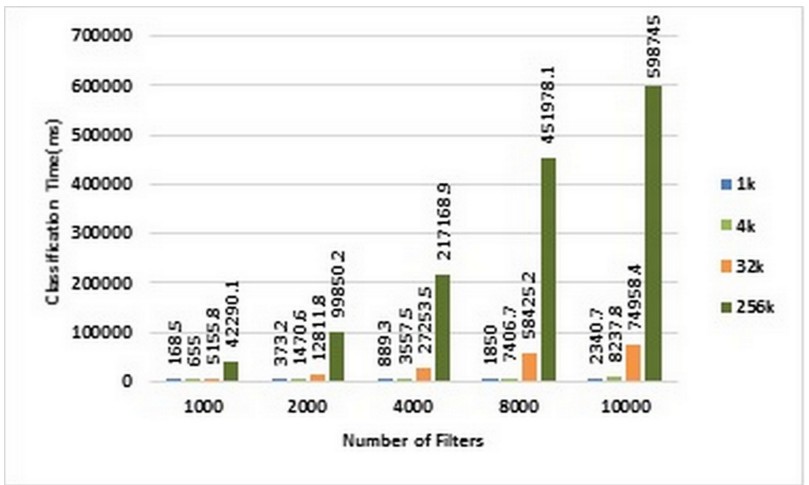

**Figure 4 Classification time in CPU.** Each bar represents the time required for classifying the specified number of packets in CPU. The number of filters is 1k, 2k, 4k, 8k, and 10k. The number of classified packets is 1k, 4k, 32k, and 256k.

graphics card. Hence, given two SMs, 4096 threads are defined. In every SM, two thread blocks with 1,024 threads are defined.

## Evaluation of results

In this section, the efficiency of the suggested mechanism for parallelization on GPU is studied from different aspects such as time of classification, throughput, and speed up. Classification time is the time required by GPU to classify all packets. Throughput is indicative of the number of packets classified per second. In this article, we have computed speedup through computing the ratio of packet classification time in the proposed kernel to the time of packet classification in the sequential version of the algorithm on CPU. The time unit for computations is millisecond and the throughput unit is kilo packets per second.

As mentioned earlier, we used ACL dataset in our experiments. Therefore, in the following, we will investigate the above benchmarks on ACL dataset. Figures 4 and 5 show the classification time of different volumes of input packets using different ACL filter sets of sizes ranging from 1K to 10k on CPU and GPU, respectively. In these two diagrams, increasing the number of input packets causes a proportional increase in classification time. For example, classification of 256K packets with 10K ACL filters on CPU lasts 598,754 ms, while it requires 97,485 ms on GPU. Therefore, GPU has been about 61.42 times faster than CPU.

For each number of filters labeled on the horizontal axis of the plot in Fig. 6, the corresponding speedups are computed using Eqs. (1) and (2) for pre-specified numbers of packets ranging from 1K to 256k on CPU and GPU, respectively. These speedups are illustrated in Fig. 6. Figure 7 illustrates corresponding speedups which have been computed using the experimental results. In each case, the experimental speedup is lower than or equal to the analytically predicted speedup. The reason is that in the analytical computation of speedup, for predicting $T_1$ and $T_P$ complexities, the worst cases are used

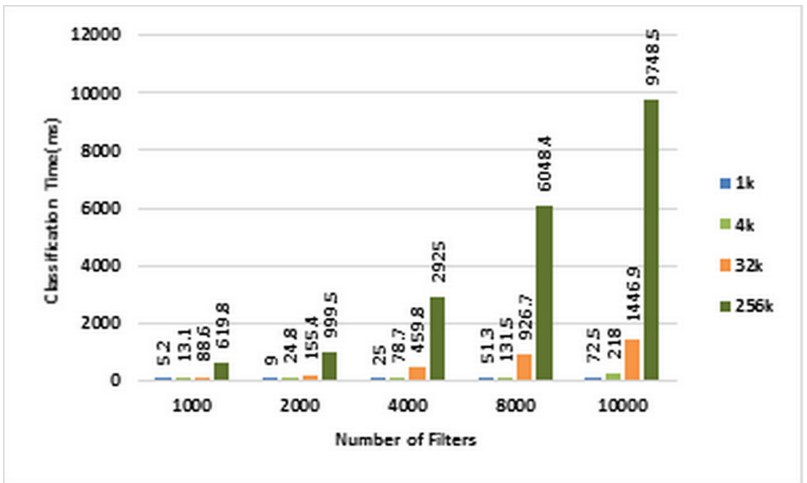

**Figure 5 Classification time in GPU.** Each bar represents the time required for classifying the specified number of packets in GPU. The number of filters is 1k, 2k, 4k, 8k, and 10k. The number of classified packets is 1k, 4k, 32k, and 256k.

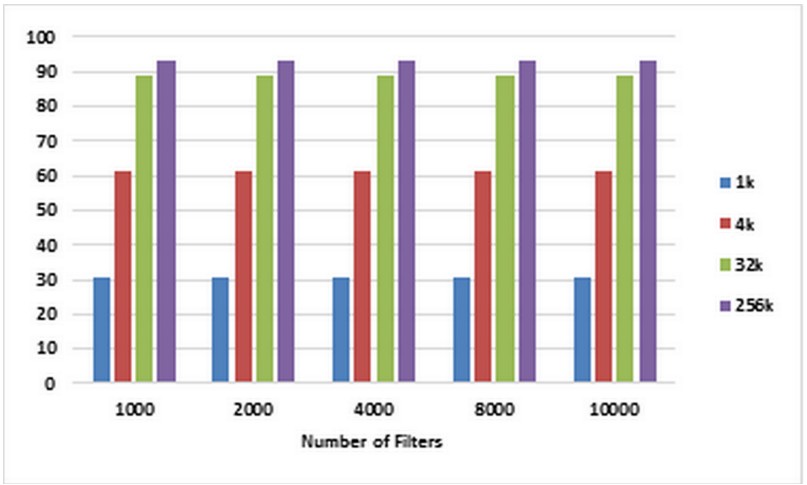

**Figure 6 Analytically predicted speedups.** Each bar represents the analytically predicted speed ups for classifying the specified number of packets. The number of filters is 1k, 2k, 4k, 8k, and 10k. The number of classified packets is 1k, 4k, 32k, and 256k.

whereas in the experimentally acquired speedup the statistical properties of packets as well as the filters can greatly affect $T_1$ and $T_P$. According to Fig. 7, in each case, by increasing the number of packets, the speedup increases. This result confirms the scalability of the parallel ABV when the number of input packets increases. The maximum speedup is achieved in the classification of 256K packets with the ABV algorithm running on a dataset with 2K filters. In this case, GPU is about 99.9 times faster than CPU.

The throughput of the sequential and parallel versions of the AVB algorithm in the classification of different numbers of packets using pre-specified numbers of filters are shown in the plots in Figs. 8 and 9, respectively. In all cases, the throughput of the parallel kernel is about 60–65 times greater than the throughput of the sequential version of ABV.

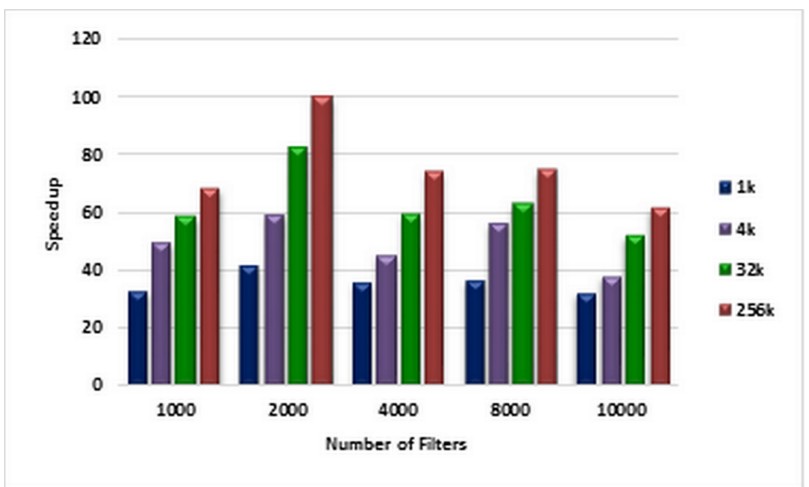

**Figure 7 Experimentally computed speedup.** Each bar represents the experimentally computed speed ups for classifying the specified number of packets. The number of filters is 1k, 2k, 4k, 8k, and 10k. The number of classified packets is 1k, 4k, 32k, and 256k.

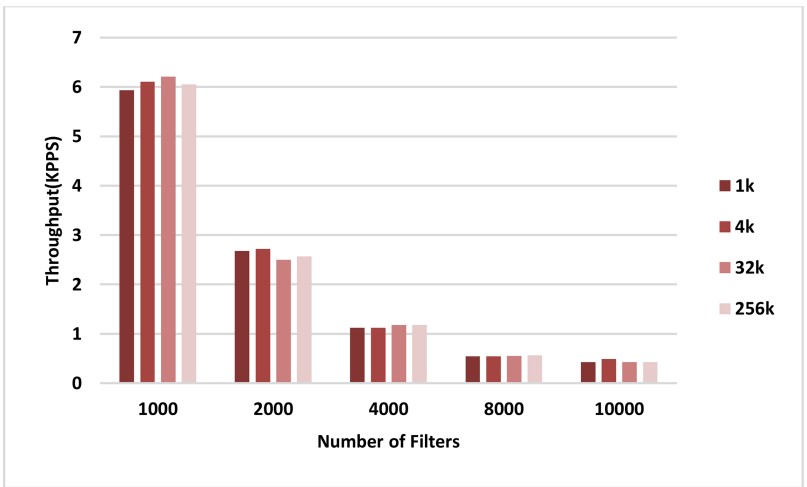

**Figure 8 CPU throughput.** Each bar represents the experimentally computed throughputs for classifying the specified number of packets in CPU. The number of filters is 1k, 2k, 4k, 8k, and 10k. The number of classified packets is 1k, 4k, 32k, and 256k.

In both figures, the throughput of packet classification is reduced by increasing the number of filters. This is due to the inevitable increase in the complexity of the ABV search, which is dependent on the number of filters. However, unlike the throughput of the sequential ABV, the throughput of the parallel kernel of ABV is increased by increasing the number of input packets. This result shows that the capability of hardware-managed threads in hiding the latency in accessing the slow global memory of GPU as well as their maximum concurrency is increased by increasing the number of input packets.

The amount of memory used for storing the ABV data structure corresponding to different sizes of ACL filter sets is displayed in the diagram of Fig. 10. Based on this diagram, the memory complexity of the ABV increases as the number of filters increases.

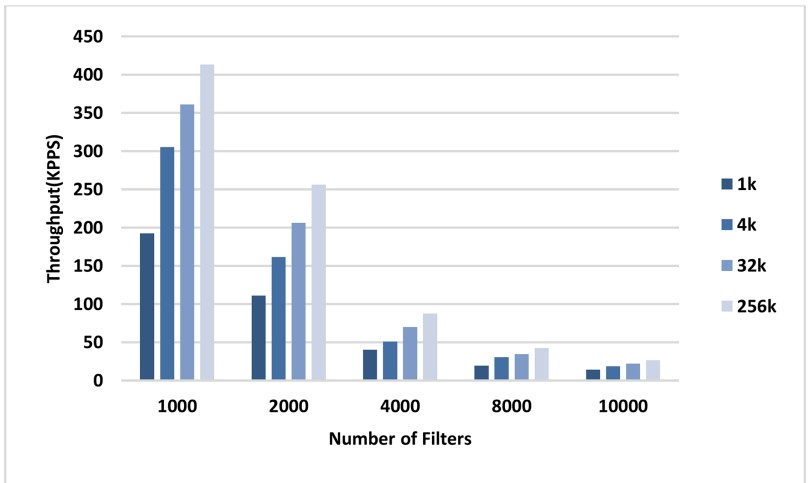

**Figure 9 GPU throughput.** Each bar represents the experimentally computed throughputs for classifying the specified number of packets in GPU. The number of filters is 1k, 2k, 4k, 8k, and 10k. The number of classified packets is 1k, 4k, 32k, and 256k.

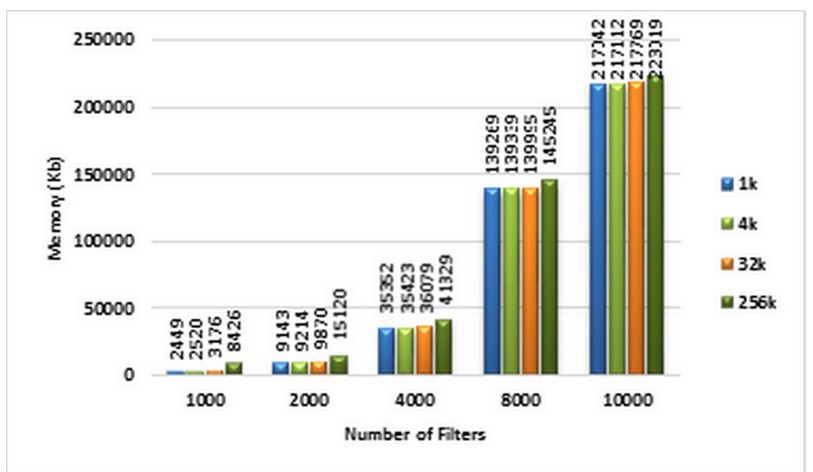

**Figure 10 Memory complexity of ABV algorithm.** Each bar represents the required memory (in Kb) for classifying the specified number of packets in GPU. The number of filters is 1k, 2k, 4k, 8k, and 10k. The number of classified packets is 1k, 4k, 32k, and 256k.

## Comparing the proposed method with recent works

In designing a parallel algorithm, it is more important to make it efficient than to make it asymptotically fast. The relative efficiency of a parallel algorithm is defined as:

$$E = \frac{S}{P} \tag{4}$$

In the above equation, $S$ and $P$ represent the highest acquired speedup and the number of processing cores on GPU, respectively. This metric reveals the maximum efficiency of a parallel algorithm on a parallel system.

The efficiency of the proposed method is compared with the efficiency of recent parallel implementations of three different packet classification algorithms (*Varvello et al., 2016*)

**Table 5 The comparison of processors efficiency in three different GPU-based packet classifications.**

| Efficiency of processing cores | The lowest acquired speedup | Number of SMs | Number of cores | Type of graphic card | Parallelized algorithm |
|---|---|---|---|---|---|
| 0.117 | 60 | 16 | 512 | GTX580 | Linear search |
| 0.195 | 100 | 16 | 512 | GTX580 | Tuple search |
| 0.214 | 110 | 16 | 512 | GTX580 | Bloom search |
| 0.257 | 99 | 2 | 384 | GT 730 | ABV |

Note:
The efficiency of the processing cores in different parallelizations of packet classification algorithms are computed and compared. Rows one to three represent the efficiency of parallel versions of Linear search, Tuple search, and Bloom search on GPU 580TX, respectively. The efficiency of parallel ABV on GT730 GPU is illustrated in row four.

in Table 5. It is obvious that the efficiency of the proposed method for the parallel classification of packets using ABV is the highest. That is, our parallel implementation of ABV, best exploits the resources of a parallel system.

## CONCLUSION

Today, packet classification has become a fundamental process in many high-speed network devices. Parallel implementation of low-complexity packet classification algorithms on GPU-like highly-threaded machines can keep the speed of this process as close as possible to the communication speed.

In this paper, we presented a method for parallelizing the ABV algorithm on GPUs using the CUDA platform. In addition, despite recent blind parallelization of packet classification algorithms, we use an analytical method which could predict the performance of the proposed method. Parallel program developers can extend our analytical framework to evaluate potential optimizations and predict the running time of any packet classification kernel without actual execution.

In order to evaluate our work, required filter sets and packets were synthetically generated using ClassBench tool. The experimental results show improvement in the performance of the parallel ABV on GPU. This improvement could be expressed as 99.9 times speedup and 65 times enhancement in the throughput. In addition, the experimental results confirm the scalability of the proposed kernel against incoming packets. Moreover, comparing the maximum efficiency of the proposed method with that of the latest implementation of three packet classification algorithms including linear, tuple space and bloom filter approves the superiority of the proposed method.

The GPU cluster has been recently considered as a high-performance platform for computation-intensive programs. Therefore, future studies should investigate how to best use such accelerators for packet classification algorithms.

### Funding

The authors received no funding for this work.

## Competing Interests

The authors declare that they have no competing interests.

## Author Contributions

- Mahdi Abbasi conceived and designed the experiments, analyzed the data, contributed reagents/materials/analysis tools, authored or reviewed drafts of the paper, approved the final draft.
- Razieh Tahouri performed the experiments, prepared figures and/or tables, performed the computation work, approved the final draft.
- Milad Rafiee performed the experiments, contributed reagents/materials/analysis tools, prepared figures and/or tables, performed the computation work, approved the final draft.

## Data Availability

The ABV kernel code is available as Supplementary Files.

## Supplemental Information

Supplemental information for this article can be found online at http://dx.doi.org/10.7717/peerj-cs.185#supplemental-information.

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
