# Peer review of "Enhancing the performance of the aggregated bit vector algorithm in network packet classification using GPU"

_PeerJ Computer Science, doi:10.7717/peerj-cs.185_

## Round 0.1 · original submission · Minor Revisions

Please make sure that all figures and text are original, or clearly attributed to their source (only after receiving the appropriate permission).

Please respond in detail to all comments by the reviewers.

Reviewer 1 ·

Basic reporting

The paper presents an implementation of a parallelized ABV algorithm on a GPU.

Language and clarity: The paper is generally written in clear English. There are some minor language issues which should be fixed (for example in lines: 40, 54, 77, 86, 132, 260, 266, 278, 301). However, the clarity of the main ideas in the paper could be improved.
For example, some parts of the text should be revised to make the text easier to follow.
1) The contributions listed in intro are not clear. Maybe provide more detail on the first contribution.
2) The background should be more clearly written. E.g., it dives right in to how the constructs of ABV work without explaining what is the purpose of the algorithm.
3) Minor issue - In the second section, I am not familiar with the use of the term instruments in this context, why not rename the section to background?

Ethics: Figure 2 is copied directly from the paper by Baboescu and Varghese, though it is not mentioned that the figure is from that paper.

Analysis: The authors present equations that represent the total time and memory of the execution. However, they do not explain the equations or prove their correctness. It may be that some of these results follow from the use of Ma et al. though that should be more rigorously exhibited and explained. In general the analysis section is hard to follow and the results are not sufficiently discussed.

Experimental design

The authors present a new way to improve the implementation of the aggregated bit vector packet classification technique presented by Baboescu and Varghese by parallelizing parts of the algorithm on GPUs.

Research question and comparison to other work: The authors present the need to speed up packet classification and propose an interesting technique. This solution should be better placed within the context of other existing solutions. There is some comparison though it is hard to infer an apples-to-apples comparison from the short comparison and table that is given.

Generalizing the result and evaluation: The discussion of the implementation and the evaluation discuss the case for two header fields. The authors should consider ways to generalize to k header fields.

Evaluation results: The purpose of the graph showing the analytic speedup is not clear, since it shows the same results for all number of filters. If the point is to compare with the results in figure 7 you may want to place the results on the same graph.
Minor issue: the legends in the graph should be labled

Validity of the findings

Data set: The authors make use of the Classbench simulator for generating files for evaluation. The toolset is publicly available making the simulations potentially reproducible, though it is no longer supported and I am not sure if it can still be used in this case.

Conclusions are provided as part of the evaluation though they should be better summarized.

Additional comments

The paper is generally well written and easy to follow, though it lacks detail of analysis and comparison to related work. Furthermore, it is not clear what would be the impact of generalizing to more fields. If that is somehow implied in the paper it is not clear.

·

Basic reporting

This paper presents a parallel implementation of aggregated bit vector algorithm for packet classification on GPU.

The paper is well written. Authors have provided sufficient literature references and background context. Experimental results are aligned with authors' analytical performance model. Terms and parameters are well defined.

Experimental design

The authors have used three major performance metrics for evaluation, including (1) classification time which is the time to classify all the packets, (2) throughput which is number of packet classified per second, and (3) speedup which is computed as the classification time of a serial CPU implementation over the classification time of the proposed parallel GPU implementation. Authors observe that the GPU implementation achieves 60-100 speedup over CPU implementation.

I suggest authors to add the performance of single packet classification latency. Because for latency-sensitive networks, this latency is more import than throughput.

Validity of the findings

The supplemental files contain the source code of the implementation and the packet files used for experiments. Those files match the authors' description in the paper.

Authors have provided the analysis of empirical results and their analytical performance models, indicating that their model can help predict accurate performance.

Additional comments

This paper is well written and structured. And it provides both empirical results and performance analysis.

I have the following suggestions:

1. The content between Line 278-281 is not reflected in Figure 3 (ABV pseudo-code). It makes me confused about how the other three fields are handled in your approach. I suggest to give more details here.

2. Table 5 seems to be an unfair comparison. This is because Varvello's CPU runs at 2.66 GHz (so their speedup number is relatively small due to a fast CPU), while your CPU runs at 2.22 GHz. A fair comparison should be based on the same baseline platform.

3. Line 399 states the memory complexity linearly increases. However, this is not true based on Figure 10. Please correct this statement.

4. Figures 8 and 9 should use KPPS as unit as stated in Line 364.

5. Fix these two sentences,
Line 75, "has an appropriate structure the lets it be highly"
Line 370, "it requires 9748/5"

---

## Round 0.2 · accepted · Accept

Please revise according to the comments of the second reviewer. Theses are minor enough that they can be resolved while in production

Reviewer 1 ·

Basic reporting

no comment

Experimental design

no comment

Validity of the findings

no comment

Additional comments

The paper has greatly improved and most of the previous concerns have been addressed.
Some comments:
1) Regarding the generality of the fields, the comment added that "For this purpose other fields of packet are inspected linearly" is not clear and still requires further explanation.
2) I do not see the needed citation in figure 2. I think it should be explicitly linked the figure in the caption or the related text about the figure.

·

Basic reporting

Overall, the paper is well written and easy to follow.
But there are a few minor reporting issues to be fixed.

Line 85-87: It is confusing here by saying 66.54 times throughput improvement corresponds to 100 times speedup. By reading evaluation session, it looks like 66.54 is average improvement number for throughput, while 100 is the maximum number for speedup?

Line 109: missing space, (Nakano 2013a)CUDA

Line 128, (Cheng et. al 2014) occurs twice

Line 408 (3) should be (4) because Line 329 already has an Equation (3)

Experimental design

Figures 6 and 7 show the comparison between predicted speedup and actual speedup. It would be better to discuss how close the predicted performance (throughput or latency, or total time) is to the measured performance as well.

Validity of the findings

No comment

Additional comments

In addition to the above minor issues, I also suggest authors to briefly discuss the motivation to select the ABV algorithm to accelerate on GPU. This discussion can be added in Introduction.